# The Novel Structural Variation in the GHR Gene Is Associated with Growth Traits in Yaks (*Bos grunniens*)

**DOI:** 10.3390/ani13050851

**Published:** 2023-02-26

**Authors:** Fubin Wang, Xiaoyun Wu, Xiaoming Ma, Qi Bao, Qingbo Zheng, Min Chu, Xian Guo, Chunnian Liang, Ping Yan

**Affiliations:** 1Key Laboratory of Yak Breeding Engineering of Gansu Province, Lanzhou Institute of Husbandry and Pharmaceutical Sciences, Chinese Academy of Agricultural Sciences, Lanzhou 730050, China; 2Key Laboratory of Animal Genetics and Breeding on Tibetan Plateau, Ministry of Agriculture and Rural Affairs, Lanzhou Institute of Husbandry and Pharmaceutical Sciences, Chinese Academy of Agricultural Sciences, Lanzhou 730050, China; 3Life Science and Engineering College, Northwest Minzu University, Lanzhou 730030, China

**Keywords:** *Bos grunniens*, growth hormone receptor (*GHR*) gene, growth traits, SV 246 bp, candidate molecular marker

## Abstract

**Simple Summary:**

Yak is the dominant animal species in China’s plateau regions. The development of the yak industry is conducive to protecting the ecosystem of the Qinghai–Tibet Plateau and promoting local economic development. However, the slow growth and development of yak have seriously affected the development of the yak industry. Structural variation (SV) has been widely applied in livestock breeding for growth traits. Therefore, it is of great significance to use SV to improve yak growth traits in yak breeding. Meanwhile, the *GHR* gene plays an important role in the formation and normal development of bones. This study associated the correlation between the yak *GHR* SV gene and growth traits and confirmed that GHR-SV can be used as a molecular marker for the early reproduction of yaks. This study provides a theoretical basis for the early growth and development of yaks.

**Abstract:**

The growth hormone receptor (*GHR*) is a member of the cytokine/hematopoietic factor receptor superfamily, which plays an important role in the growth and development, immunity, and metabolism of animals. This study identified a 246 bp deletion variant in the intronic region of the *GHR* gene, and three genotypes, including type II, type ID, and type DD, were observed. Genotype analysis of structural variation (SV) was performed on 585 individuals from 14 yak breeds, and it was found that 246 bp deletion was present in each breed. The II genotype was dominant in all yak breeds except for SB yak. The association analysis of gene polymorphisms and growth traits in the ASD yak population showed that the 246 bp SV was significantly associated with body length at 6 months (*p* < 0.05). *GHR* messenger RNA (mRNA) was expressed in all the tested tissues, with significantly higher levels in the liver, muscle, and fat than in other organs. The results of transcription activity showed that the luciferase activity of the pGL4.10-DD vector was significantly higher than that of the pGL4.10-II vector (*p <* 0.05). Additionally, the transcription-factor binding prediction results showed that the SV in the runt-related transcription factor 1 (Runx1) transcription-factor binding site may affect the transcriptional activity of the *GHR* gene, regulating yak growth and development. This study showed that the novel SV of the *GHR* gene could be used as a candidate molecular marker for the selection of the early growth trait in ASD yak.

## 1. Introduction

Yak (*Bos grunniens*) mainly lives in the Qinghai–Tibet Plateau at an altitude of 3000–5500 m [1]. In China, there are about 16 million yaks, accounting for about 95% of the world’s population [2]. It is an important and dominant animal species in plateau animal husbandry, which can provide meat, milk, fur, etc. [3]. Among them, yak meat is one of the main resources of the plateau animal husbandry economy [4]. Compared with cattle beef, yak meat is high in protein and energy, low in fat, and rich in amino acids, and the habitat of yak is least affected by humans. Therefore, yak meat is considered a natural green food, which meets people’s demand for green and healthy meat quality [5]. As a plateau variety, yak can protect the Qinghai–Tibet Plateau ecosystem and promote local economic development. Compared with cattle, the growth rate of yak is slow due to the lack of an efficient yak breeding program to improve growth traits, which affects the yield of yak meat. Therefore, it is of great significance to study the growth traits of yak.

The growth hormone receptor (*GHR*) gene is an essential growth-related candidate gene that plays a critical role in the early stages of animal growth and development [6]. The growth hormone (GH)–insulin-like growth factor (IGF) growth axis is crucial for the growth and differentiation of skeletal muscle [7]. The *GHR* gene combines with GH to initiate the intracellular signal transduction mechanism, increase the expression of insulin-like growth factor 1 (IGF-1), promote cell proliferation, and affect the growth and development of animal skeletal muscle [8]. In mice, *GHR* knockouts exhibit impaired skeletal muscle development characterized by a reduced number and area of muscle fibers and concomitant functional defects. *GHR* gene variants are associated with growth and development in livestock. The *GHR* gene is considered to be a molecular genetic marker of growth performance in fertile pigs [9]. In sheep, the *GHR* gene variation is associated with growth traits [10]. According to recent studies, the presence of genetic variations in the *GHR* gene is associated with economic traits, such as cattle growth and production [11,12]. These studies suggest that genetic variation in the *GHR* gene plays an important role in regulating animal growth and development.

Structural variation (SV) is an important and abundant source of genetic and phenotypic variation, including insertions/deletions (Indel), duplications, translocations, and copy-number variations (CNVs) [13]. Compared with a single nucleotide variant (SNV), SVs could have a greater impact on the function and expression of genes [14]. Up to now, many SVs are found to be associated with economic traits in domestic animals. For example, a deletion of 110 kb in the *MER1* repeat containing the imprinted transcript 1 (*MIMT1*) gene is associated with abortion and stillbirth in cattle [15]. The lysine acetyltransferase 6A (*KAT6A*) gene is associated with the development of different body size traits in sheep [16]. In addition, a 225 bp deletion in the glutaminyl–peptide cyclotransferase-like (*QPCTL*) gene was associated with body weight and carcass traits in chickens [17]. Upadhyay et al. [18] detected an SV in the regulatory region of polypeptide N-acetylgalactosaminyltransferase-like 6 (*GALNTL6*) associated with feed efficiency and growth traits in cattle. However, there is no detailed study on SVs of the yak *GHR* gene.

In this study, we found a 246 bp sequence deletion SV of the yak *GHR* gene. In order to identify the role of the *GHR* gene SV in yak populations in China, we analyzed the association between *GHR* 246 bp deletion variation and phenotypic traits to illustrate the genetic effect of *GHR* 246 bp deletion variation in yak breeds. This study provides useful information for the genetic protection and improvement of Chinese yak.

## 2. Materials and Methods

### 2.1. Ethics Statement

This work has been approved by the Lanzhou Institute of Husbandry and Pharmaceutical Sciences; the grant number is No. LIHPS-CAAS-2017-115. In addition, we collected all blood samples and analyzed the data strictly following the guidelines for the Care and Use of Laboratory Animals.

### 2.2. Sample Collection

Blood samples were collected from a total of 585 individuals of 14 yak breeds. The numbers of samples from each breed were as follows: Datong yak (DT, n = 22); Xueduo yak (XD, n = 21); Huanhu yak (HH, n = 21); Gannan yak (GN, n = 21); Tianzhu White yak (TZB, n =21); Niangya yak (NY, n = 18); Leiwuqi yak (LWQ, n = 21); Sibu yak (SB, n = 18); Muli yak (ML, n = 23); Jiulong yak (JL, n = 21); Maiwa yak (MW, n = 21); Zhongdian yak (ZD, n = 21); Bazhou yak (BZ, n = 21); and Ashidan yak (ASD, n = 315). We tracked the phenotypic data of these Ashidan yaks (i.e., body weight (BW, kg), withers height (WH, cm), body length (BL, cm), and chest girth (CG, cm)) at four age groups of growth (i.e., 6 months old, 12 months old, 18 months old, and 30 months old). The phenotype was measured using the standard method of measurement [19]. Tissues, including the heart, muscle, liver, spleen, kidney, lung, and brain, were collected for expression profiling.

### 2.3. Genomic DNA (gDNA) Extraction and Polymerase Chain Reaction (PCR)

Yak blood gDNA was extracted using the Tiangen whole-genome blood extraction kit (Beijing, China), and the DNA quality was detected using an ultra-microspectrophotometer (Thermo Fisher Scientific, USA). The gDNA was stored at −20 °C for further analysis. The primers for gene structure variation were designed by the National Center for Biotechnology Information (NCBI) online primer design software Primer-BLAST and synthesized by Qingke Zexi (Xi’an) Biotechnology company (Xi’an, Shaanxi, China) (Table 1). PCR was then performed. A total volume of 10 μL PCR mixture contained 1 μL of gDNA (50 ng/μL), 1 μL of each primer (10 μmol/L), 5 μL of Go Taq^®^Green Master Mix (Promega, Madison, WI, USA), and 2 μL of ddH_2_O. The PCR conditions were as follows: 95 °C for 3 min, 95 °C for 5 s, 58 °C for 30 s, 72 °C for 1 min for 30 cycles, and 72 °C for 5 min. The products were identified via 2% agarose gel electrophoresis. Six PCR products were selected and sequenced by Qingke Zexi (Xi’an) Biotechnology Company (Xi’an, Shaanxi, China).

### 2.4. RNA Extraction and Quantitative PCR Identification

The total RNA of yak tissue was extracted using the Trizol method, and the integrity of RNA was identified with 1% agarose gel electrophoresis. cDNA was synthesized via reverse transcription using a translator first-strand cDNA synthesis kit (Roche, Shanghai, China) and stored at −20 °C for further analysis. Quantitative real-time polymerase chain reaction (RT-qPCR) was performed using Go Taq^®^qPCR and RT-qPCR Systems (Promega, Madison, WI, USA) to investigate the expression levels of the GHR messenger RNA (mRNA) in each tissue. Three replicates were selected for each sample. The primer information is provided in Appendix A. The RT-qPCR conditions were as follows: 95 °C for 3 min, 95 °C for 5 s, 64 °C for 20 s for a total of 40 cycles, and 72 °C for 30 s. The results were analyzed using the 2^−ΔΔCT^ method [20].

### 2.5. Cell Culture and Cell Transfection

The 293T cell line was purchased from the Cell Bank of the Chinese Academy of Sciences (Shanghai, China). High-sugar DMEM (Hyclone, Logan, UT, USA), fetal bovine serum (Gibco, Waltham, MA, USA), and 1% streptomycin/penicillin (Invitrogen, Waltham, MA, USA) were used. Plasmids were transfected into cells using ViaFectTMTransfection Reagent (Promega, Madison, WI, USA) according to the manufacturer’s instructions.

### 2.6. Plasmid Construction and Prediction of Transcription-Factor Binding Sites (TFBSs)

Based on the yak *GHR* 246 bp deletion locus, the luciferase reporter vector pGL4.10 was selected, and the pGL4.10-II (324 bp, containing the 246 bp deletion) and pGL4.10-DD (78 bp, lacking the 246 bp deletion) vectors were constructed to detect the transcriptional activity of *GHR* 246 bp deletion.

In this study, the online software AnimalTFDB 3.0 (http://bioinfo.life.hust.edu.cn/AnimalTFDB/#!/) (accessed on 10 February 2022) and JASPAR (http://jaspar.genereg.net/) (accessed on 10 February 2022) were used for the prediction of TFBSs at the *GHR* 246 bp deletion locus. The runt-related transcription factor 1 (Runx1) binding sequence (TAAATGCAAA) was identified on the II genotype, and the pGL4.10-KO-Runx1 dual-luciferase reporter vector (324 bp) was constructed. The vector pGL4.75 (Promega, Madison, WI, USA) was used as an internal reference vector for determining the luciferase reporter system.

### 2.7. Statistical Analysis

Gene frequency and genotype frequency were calculated using the following equations:

Genotype frequency (FAiAj) = AiAj number of individuals/total number of samples;
Allele frequency (FAi) = FAiAj + 1/2 FAiAj.

The online website SHEsis was used to calculate Hardy–Weinberg equilibrium, heterozygosity, polymorphism information, and genetic variation content of *GHR* gene SV in the population. Meanwhile, we analyzed the association between the SV type of the *GHR* gene and the growth traits of Ashidan yak by using a one-way analysis of variance (ANOVA) and frequency-distribution histogram test. A mixed linear model used in the analysis is as follows: Yij = μ + Gij + eij, where Yij is the observed growth trait; μ is the population average of each trait; Gij is the fixed effect of SV genotype of the *GHR* gene; and eij is the random residual. The difference between the mean values was evaluated by using one-way ANOVA with a post-event least significant digit (LSD) multiple-comparison test. The software used for the statistical analyses was the IBM SPSS Statistics software (Version 23.0). Data are expressed as mean± standard error (SE).

## 3. Results

### 3.1. Identification of SV of Yak GHR Gene

A 246 bp deletion variation was identified in intron 6 of the *GHR* gene (NW_005393449.1: 3052434-3052680). Based on the designed primers (F1, R1), GHR-SV246 was amplified with PCR using yak gDNA as the template. The GHR-SV246 polymorphism was analyzed via PCR amplification and agarose gel electrophoresis of the product (Figure 1). The product length matched the expectation, and three genotypes were identified, namely II (778 bp), ID (778 bp and 532 bp), and DD (532 bp). The sequencing analysis of the amplified products showed that the sequencing results were consistent with the reference genome sequence.

### 3.2. Identification of SV of Yak GHR Gene

The genotype frequencies and allele frequencies of GHR-SV246 in different yak breeds are shown in Table 2. The frequencies of different genotypes in Chinese yak populations were 0.723, 0.202, and 0.075, respectively. The II genotype was the dominant genotype in different populations except for SB yak; the allele frequencies were 0.824 and 0.176, respectively. Among them, the frequency of the I allele was the highest in the ML yak population (0.979) and the lowest in the SB yak population (0.500) (Table 2). The Ho ranged from 0.500 to 0.959, indicating high heterozygosity within different yak breeds, and the Ne ranged from 1.04 to 2.00. Classification based on the polymorphism information content (PIC value < 0.25, low polymorphism; 0.25 < PIC value < 0.5, medium polymorphism; PIC value > 0.5, high polymorphism) showed that the 246 bp deletion of the GHR gene exhibited relatively low polymorphism in ZD, ML, LWQ, NN, GN, and XD yak populations, while moderate polymorphism was observed in other yak populations (Table 3).

### 3.3. The Association of the 246 bp Deletion of GHR with Growth Traits of Yak

In this study, the normal distribution of yak growth traits (Appendix A) and the average value of the growth parameters were statistically analyzed (Appendix A). We also analyzed the association of *GHR* gene polymorphisms with the growth traits in yaks using a linear model. As shown in Table 4, the polymorphisms in the *GHR* gene were significantly associated with body length in 6-month-old ASD yaks (*p* < 0.05), while there was no significant relationship with other growth traits (*p* > 0.05).

### 3.4. Identification of Fluorescence Activity of GHR Gene Polymorphism in Yak

To detect the transcriptional activity of different GHR-SV246 genotypes in cells, pGL4.10, pGL4.10-II, and pGL4.10-DD vectors were co-transfected into 293T cells with the internal reference plasmid pGL4.75. The fluorescence activity of the pGL4.10-II vector was significantly different from that of the blank control group pGL4.10 (*p* < 0.05) and was significantly lower than that of the pGL4.10-DD vector (*p* < 0.05) (Figure 2).

### 3.5. Effect of Transcription Factors on the Transcriptional Activity of Mutant Recombinant Plasmids

In order to determine the effect of Runx1 transcription-factor binding with the 246 bp deletion on transcriptional activity in cells, the pGL4.10-KO-Runx1 luciferase reporter vector with mutant Runx1 binding site was constructed, and its fluorescence activity was identified. The luciferase activity of vector pGL4.10-KO-Runx1 was significantly lower than that of vector pGL4.10-II (Figure 3; *p* < 0.05), indicating that the transcription factor Runx1 could be combined with the II genotype sequence, thus increasing its transcriptional activity in cells.

### 3.6. mRNA Expression Profile of the GHR Gene in Yak

The expression levels of the *GHR* gene in different tissues were identified in yaks. There were significant differences in the expression of the *GHR* gene in various tissues of yaks, with the highest expression in fat, followed by the liver, muscle, and kidney tissues, and the lowest expression in the heart, spleen, and lung tissues (Figure 4).

## 4. Discussion

SV is a major determinant of the phenotypic diversity of animals. Studies on chickens [21], sheep [22], and cattle [10] have reported that SVs play a significant role in their genetic diversity. In poultry, the SVs of prolactin receptor (*PRLR*) [23] and *Lpin1* [24] were significantly associated with chicken growth, carcass characteristics, and other economic traits. In cattle, the SVs of *SIRT4* and *NPM1* were significantly associated with bovine growth traits and meat quality [25,26]. These studies show that SV is an important class of molecular genetic markers, which is of great value in revealing the genetic mechanism of the economic traits of livestock and poultry.

The *GHR* gene is a member of the cytokine receptor superfamily [27]. Some studies have pointed out that the *GHR* gene can regulate the growth of skeletal muscle by interacting with the growth hormone (*GH*) [22,28]. In this study, we analyzed the expression of the *GHR* gene in different tissues of yak and found that the *GHR* gene was expressed in different tissues, with high expression in adipose tissue and moderate expression in the muscle. This is consistent with previous studies on *GHR* expression profiles [29]. *GHR* mRNA expression levels in muscle and fat have a positive effect on post-weaning weight gain in cattle, while *GHR* mRNA expression in the liver has a negative effect on post-weaning weight gain in cattle [30]. A study reported that there is a natural antisense transcript (GHR-AS) on the *GHR* gene of chickens, and its overexpression can promote the expression of myogenic differentiation 1 (*MyoD*) and myosin heavy chain (*MyHC*) and the differentiation of myoblasts, thus enhancing the differentiation of myoblasts [31].

A previous study found associations between *GHR* gene polymorphisms and the growth traits in cattle at different ages [16], showing that *GHR* gene variation can affect the growth and production traits of cattle. However, few studies have reported an association between *GHR* gene variation and yak growth and production traits. In this study, we found a 246 bp structural variation site in intron 6 of the yak *GHR* gene. The genotyping results showed that the three genotypes of *GHR* genes existed in different yak breeds, and the analysis of genetic parameters suggested that the 246 bp deletion was in Hardy–Weinberg equilibrium in different yak breeds, which indicates that it is relatively stable in yak population. In ASD yak, this SV belongs to a moderate polymorphism, indicating that the degree of genetic variation is relatively large, and it has certain breeding potential. We found that individuals with the DD genotype showed higher body length at 6 months of age in ASD yak breeds. These data suggest that SVs in the *GHR* gene might influence the early development of yak.

Introns are noncoding segments of genes that are removed via splicing during gene transcription and ultimately do not exist in mature RNA molecules [32]. Introns play an important role in the regulation of gene expression networks. Ostrovsky et al. [33] found that the intron 9 of the heparanase (*HPSE*) gene has a regulatory effect on its expression, and the existence of mutation sites in this intron has a significant impact on the regulatory effect of the intron. In this study, the transcriptional activity of the deletion sequence in the yak *GHR* gene was analyzed, and the luciferase activity of the pGL4.10-DD vector was significantly higher than that of the pGL4.10-II vector (*p* < 0.05). This finding indicates that there may be repressive TFBSs in the II genotype sequence, which interferes with the expression of the *GHR* gene by binding to transcription factors and affects the growth and development of yaks. Genomic variation in introns can affect gene expression through cis-element targets or by binding to other functional genes [27]. Boriushkin et al. [34] found a Krüppel-like factor 4 (KLF4) transcription-factor binding site located in the intron of the delta-like canonical Notch ligand 4 (*DLL4*) gene, which is a key regulator of angiogenesis. They also found that KLF4 can inhibit the expression of *DLL4* via this binding site. The appearance of gene variant sites may introduce novel transcription-factor binding sites or destroy existing binding sites [35,36]. In this study, in the II genotype sequence, we detected a Runx1 transcription-factor binding site, which was first discovered in acute myeloid leukemia [37]. Runx1 has an important regulatory role in the growth and development of animal muscles. In mouse muscle cells, the overexpression of Runx1 inhibits myogenic differentiation but promotes myoblast proliferation [38]. When skeletal muscle is injured, the expression of *Runx1* is upregulated in skeletal muscle, thereby inhibiting the premature differentiation of primary myoblasts and promoting muscle regeneration [39]. Runx1 inhibits *Spi1* gene expression by interacting with *Spil* introns [29]. In this study, we hypothesized that Runx1 has an effect on the transcriptional regulation of the *GHR* gene II genotype in yaks. This study found that changes in Runx1-TFBS reduced the fluorescence activity of the II genotype vector, indicating that the Runx1 could bind to the II genotype sequence and increase the transcriptional activity of the *GHR* gene. The luciferase activity of the pGL4.10-II vector was not significantly different from the pGL4.10 empty-vector fluorescence activity (*p* > 0.05). There may also be a binding site of an inhibitory transcription factor in the II genotype sequence.

## 5. Conclusions

We detected and validated the SV of the *GHR* gene in yaks for the first time. The results indicated that SV of the *GHR* gene was significantly correlated with body length at 6 months of age in ASD yaks. Moreover, the transcriptional activity assay found significant differences among different genotypes, and transcription factor Runx1 promoted transcriptional activity in the II genotype. Our study provides primary evidence for the role of the *GHR* gene, which may be a molecular marker for early yak breeding in the future.

## Figures and Tables

**Figure 1 animals-13-00851-f001:**
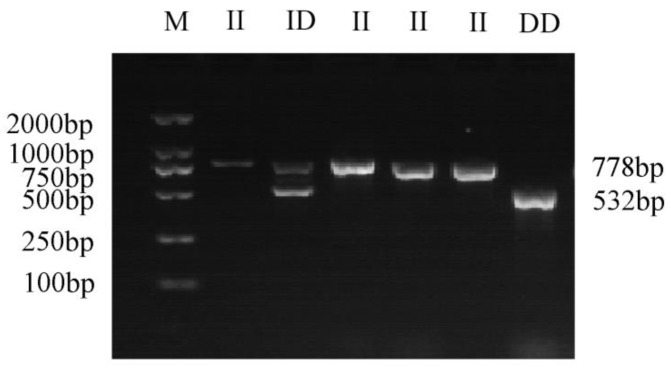
The gel electrophoresis of different genotypes of GHR-SV246; M, GL DNA marker 2000; II, homozygous insertion type; ID, heterozygous type; DD, homozygous deletion type.

**Figure 2 animals-13-00851-f002:**
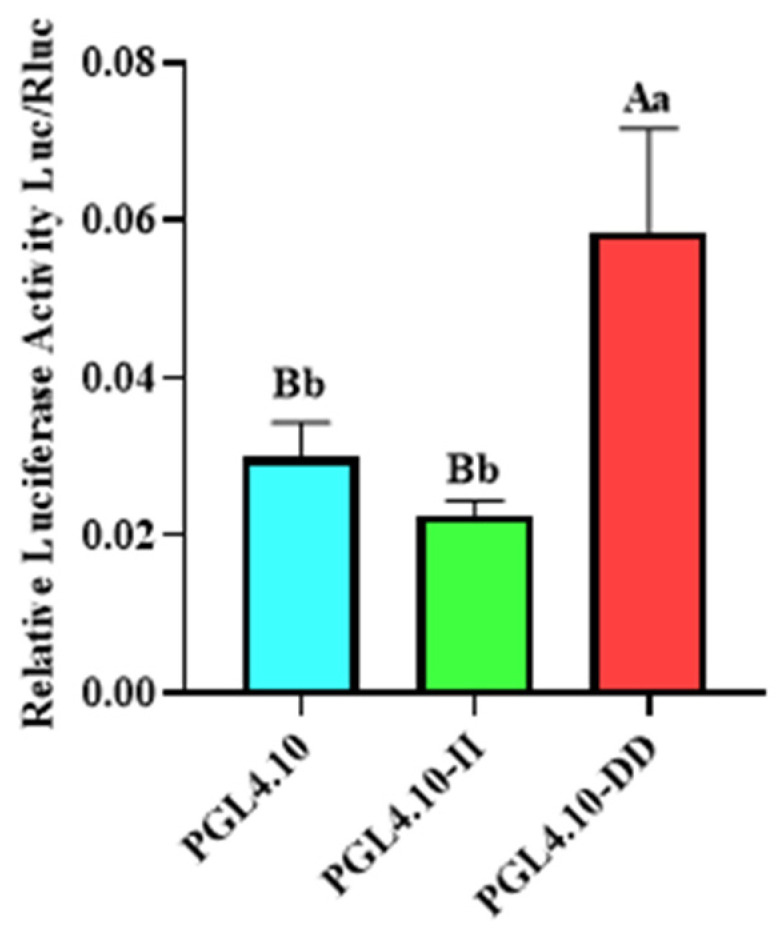
The dual-luciferase activity of GHR-SV246 mutant recombinant plasmid in 293T cells; the same letter means no significant difference (*p* > 0.05); different letters indicate significant differences (*p* < 0.05).

**Figure 3 animals-13-00851-f003:**
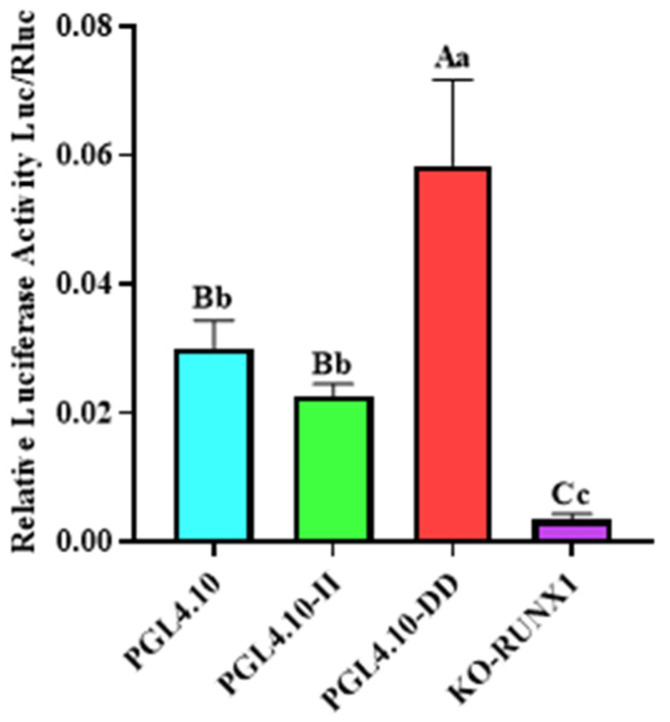
The dual-luciferase activity of GHR-SV246 recombinant plasmid in 293T cells.

**Figure 4 animals-13-00851-f004:**
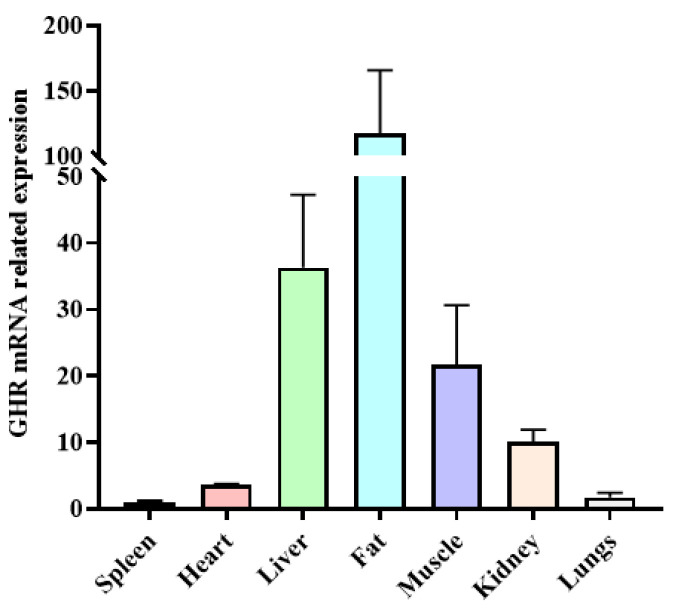
The expression level of *GHR* gene mRNA in different tissues of yak.

**Table 1 animals-13-00851-t001:** The information of primer sequence.

Levels	Gene	Primer Sequence (5′-3′)		Product Length (bp)	Tm (°C)
DNA	*GHR*	F1	TCAGAGATGAGCAACAGTGCC	778	58.0
		R1	TGCGTATCTACACCTGAGCAC		
		F2	GGggtaccTGAGCAACAGTGCCCCATTT	324	64.9
		R2	GAagatctTCACACACTCTAGACCTTAAAGCTG		
RNA		F3	CAGCAGCCCAGTGTTATCCT	230	64.0
		R3	AATGTCGCTTACCTGGGCAT		
	*β-actin*	F4	GCAGGTCATCACCATCGG	177	64.0
		R4	CCGTGTTGGCGTAGAGGT		

F: forward primer; R: reverse primer.

**Table 2 animals-13-00851-t002:** Genotype and allele frequencies of GHR-SV246 in different Yaks.

Breeds	Sample Size/n	Genotype Frequency			Allele Frequency		Description
		II	ID	DD	I	D	
Ashidan yak (ASD)	315	0.714	0.181	0.105	0.805	0.195	Domestic, Datong Qianghai, PRC
Datong yak (DT)	22	0.636	0.364	0	0.818	0.182	Domestic, Datong Qianghai, PRC
Xueduo yak (XD)	21	0.762	0.238	0	0.881	0.119	Domestic, Henan Qianghai, PRC
Huanhu yak (HH)	21	0.524	0.286	0.190	0.667	0.333	Domestic, Haibei Qianghai, PRC
Gannan yak (GN)	21	0.950	0.0500	0	0.977	0.0230	Domestic, Gannan Gansu, PRC
Tianzhubai yak (TZB)	21	0.666	0.286	0.0480	0.810	0.190	Domestic, Tianzhu Gansu, PRC
Niangya yak (NY)	18	0.889	0.111	0	0.944	0.0560	Domestic, Naqu Tibet, PRC
Leiwuqi yak (LWQ)	21	0.714	0.286	0	0.857	0.143	Domestic, Changdu Tibet, PRC
Sibu yak (SB)	18	0.500	0.500	0	0.500	0.500	Domestic, Sibu Tibet, PRC
Muli yak (ML)	23	0.957	0.0430	0	0.979	0.0210	Domestic, Liangshan Sichuan, PRC
Jiulong yak (JL)	21	0.667	0.0950	0.238	0.714	0.286	Domestic, Jiulong Sichuan, PRC
Maiwa yak (MW)	21	0.500	0.0910	0.409	0.545	0.455	Domestic, Hongyuan Sichuan, PRC
Bazhou (BZ)	21	0.667	0.238	0.0950	0.786	0.214	Domestic, Hejing Xinjiang, PRC
Zhongdian yak (ZD)	21	0.571	0.333	0.0960	0.738	0.262	Domestic, Zhongdian Yunnan, PRC
Total	586	0.723	0.202	0.0750	0.824	0.176	

II, homozygous insertion type; ID, heterozygous; DD, homozygous deletion type.

**Table 3 animals-13-00851-t003:** The genetic parameters of Yak GHR-SV246.

Breeds	Sample Size/n	HWE *p*-Value	Genetic Parameters				Description
			Ho	He	Ne	PIC	
Ashidan yak (ASD)	315	0.493	0.686	0.313	1.45	0.265	Domestic, Datong Qianghai, PRC
Datong yak (DT)	22	0.474	0.702	0.298	1.42	0.253	Domestic, Datong Qianghai, PRC
Xueduo yak (XD)	21	0.365	0.790	0.210	1.27	0.187	Domestic, Henan Qianghai, PRC
Huanhu yak (HH)	21	0.636	0.556	0.444	1.80	0.346	Domestic, Haibei Qianghai, PRC
Gannan yak (GN)	21	0.109	0.955	0.0449	1.04	0.0439	Domestic, Gannan Gansu, PRC
Tianzhubai yak (TZB)	21	0.486	0.692	0.308	1.44	0.260	Domestic, Tianzhu Gansu, PRC
Niangya yak (NY)	18	0.216	0.894	0.106	1.12	0.100	Domestic, Naqu Tibet, PRC
Leiwuqi yak (LWQ)	21	0.410	0.755	0.245	1.32	0.215	Domestic, Changdu Tibet, PRC
Sibu yak (SB)	18	0.693	0.500	0.500	2.00	0.375	Domestic, Sibu Tibet, PRC
Muli yak (ML)	23	0.102	0.959	0.0411	1.04	0.0403	Domestic, Liangshan Sichuan, PRC
Jiulong yak (JL)	21	0.599	0.592	0.408	1.69	0.325	Domestic, Jiulong Sichuan, PRC
Maiwa yak (MW)	21	0.689	0.504	0.496	1.98	0.373	Domestic, Hongyuan Sichuan, PRC
Bazhou (BZ)	21	0.519	0.664	0.336	1.51	0.280	Domestic, Hejing Xinjiang, PRC
Zhongdian yak (ZD)	21	0.575	0.613	0.387	1.63	0.312	Domestic, Zhongdian Yunnan, PRC

HWE, Hardy–Weinberg equilibrium; Ho, homozygosity; He, heterozygosity; Ne, number of effective alleles; PIC, polymorphic information content.

**Table 4 animals-13-00851-t004:** The association analysis between different body size traits and different *GHR* genotypes of Yaks.

Age	Growth Trait	SV Types			*p*-Value
		II (0.714)	ID (0.181)	DD (0.105)	
6 months (n = 315)	Body weight	84.711 ± 0.692	83.754 ± 1.374	83.000 ± 1.806	0.601
	Body length	91.667 ± 0.485 ^b^	91.386 ± 0.963 ^b^	95.242 ± 1.266 ^a^	<0.05 *
	Body height	94.356 ± 0.350	94.684 ± 0.695	94.758 ± 0.914	0.861
	Chest girth	123.938 ± 0.513	124.175 ± 1.020	124.364 ± 1.341	0.944
12 months (n = 315)	Body weight	82.698 ± 0.690	81.140 ± 1.389	83.970 ± 1.826	0.433
	Body length	95.920 ± 0.332	95.807 ± 0.659	97.182 ± 0.866	0.370
	Body height	90.502 ± 0.281	90.561 ± 0.558	91.121 ± 0.734	0.733
	Chest girth	117.187 ± 0.332	117.281 ± 0.660	117.970 ± 0.867	0.701
18 months (n = 226)	Body weight	122.329 ± 1.023	123.550 ± 2.014	120.548 ± 2.288	0.616
	Body length	101.335 ± 0.455	102.600 ± 0.895	101.581 ± 1.017	0.454
	Body height	102.903 ± 0.459	102.900 ± 0.903	102.258 ± 1.026	0.843
	Chest girth	138.387 ± 0.827	142.350 ± 1.628	138.129 ± 1.849	0.084
30 months (n = 180)	Body weight	154.944 ± 1.121	155.375 ± 2.621	159.273 ± 3.161	0.451
	Body length	113.421 ± 0.509	112.906 ± 1.009	113.136 ± 1.217	0.893
	Body height	99.286 ± 0.450	99.125 ± 0.892	100.955 ± 1.076	0.322
	Chest girth	146.484 ± 0.730	147.125 ± 1.448	146.818 ± 1.746	0.919

^a,b^ Values represent statistically significant differences at a level of *p* < 0.05; * values represent significant differences at a level of *p* < 0.05.

## Data Availability

This study did not report any data.

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
