# Peer review of "The Novel Structural Variation in the GHR Gene Is Associated with Growth Traits in Yaks (Bos grunniens)"

_animals, 2023, doi:10.3390/ani13050851_

Round 1

Reviewer 2 Report

The detection and validated of the SV of the GHR gene in yaks is a fact! Congratulation!

It the abstract - Acronim "SB" yak have to be explanation in the resume. Also "ASD" yak.

The acronyms from figure have to be more described in figures' legend (like those: pGL4.10, pGL4.10-II, and pGL4.10-DD...in figure 2 and all those + e pGL4.10-KO-Runx1 in figure 3). 

Significance values (p < 0.05) in the conclusion are not need it. 

Reviewer 3 Report

I feel that this manuscript is appropriate to be published in the journal. I have some suggestions regarding minor language issues that I listed along with the manuscript.

I have two questions. Is there any effect of deletion on the structure of mRNA? Are there any other meaningful features in the deleted sequence?

Reviewer 4 Report

This manuscript investigated “The Novel Structural Variation of GHR Gene is Associated  with Growth Traits in Yaks (Bos grunniens)”. The content is fall into the scope of the present journal. The topic is interest, and the manuscript also raised many concerns.

Please review the statistical analysis with a specialist in this field, as there are gross errors in the statistical analysis that would cast doubt on the accuracy of the obtained results.

Are you tested the normal distribution of growth data? If yes… please add that in the statistical analysis in M&M part.

What is the name of the test that you used to examine the significant differences between means?

You referred to the use of the mixed model that includes the influence of age and genotype. I think that the interaction between them should be studied, with the addition of other fixed effects such as season of birth, birth weight, and other variables related to the yak dams, such as parity, age at first birth….. etc.

Please add a table includes the overall average of all considered growth parameters

The output of the statistical model used does not agree with the tabulated data in Table 4... The output of this model will show the effect of genotype and the effect of age, while the tabulated data is an interaction between them, although it is not included in the statistical model.

The data shown in Figures 4 and 5, I think it is not subject to normal distribution, and in this case, you must use the Kruskal-Wallis test. Did you use this test or not?

 Are all Yaks bred in one place or in multiple areas?

There are multiple indicators of growth. Why did you choose only the four indicators included in the research, even though they are not significant among the genotypes?

Round 2

Reviewer 4 Report

You did not correct the modifications that I mentioned earlier, related to the statistical analysis. Please revise the statistical analysis with specialists in this field.

Author Response

Thank you for your suggestions, which are very helpful for the writing of the paper. Please see the attachment.
